# Microbiological Aspects and Enzymatic Characterization of *Curvularia kusanoi* L7: Ascomycete with Great Biomass Degradation Potentialities

**DOI:** 10.3390/jof10120807

**Published:** 2024-11-21

**Authors:** Maryen Alberto Vazquez, Luis Rodrigo Saa, Elaine Valiño, Livio Torta, Vito Armando Laudicina

**Affiliations:** 1Institute of Animal Science (ICA), San José de Las Lajas 32700, Cuba; elainevalino@gmail.com; 2Laboratorio de Sanidad Animal y Zoonosis, Departmento de Ciencias Biologicas y Agropecuria, Escuela de Ingeniería Agropecuaria, Universidad Técnica Particular de Loja, Loja 1101608, Ecuador; 3Dipartmento di Scienze Agrarie, Alimentari e Forestali, Università degli Studi di Palermo, 90133 Palermo, Italy; livio.torta@unipa.it (L.T.); vitoarmando.laudicina@unipa.it (V.A.L.)

**Keywords:** fungi, cell wall degrading enzymes, carbon mineralization, fiber modification

## Abstract

The complex structure of the plant cell wall makes it difficult to use the biomass produced by biosynthesis. For this reason, the search for new strains of microorganisms capable of efficiently degrading fiber is a topic of interest. For these reasons, the present study aimed to evaluate both the microbiological and enzymatic characteristics of the fungus *Curvularia kusanoi* L7strain. For this, its growth in different culture media was evaluated. Wheat straw mineralization was evaluated by gas chromatography assisted by infrared spectroscopy. The production of endo- and exoglucanase, laccase, and peroxidase enzymes in submerged solid fermentation of wheat and sugarcane bagasse were characterized. The strain efficiently mineralized raw wheat straw, showing a significant decrease in signals associated with cellulose, hemicellulose, and lignin in the infrared spectra. High enzyme productions were achieved in submerged solid fermentation of both substrates, highlighting the high production of endoglucanases in sugarcane bagasse (2.87 IU/mL) and laccases in wheat (1.64 IU/mL). It is concluded that *C. kusanoi* L7 is an ascomycete with a versatile enzyme production that allows it to exhaustively degrade complex fibers such as raw wheat straw and sugar cane bagasse, making it a microorganism with great potential in the bioconversion of plant biomass.

## 1. Introduction

The cell wall of plants has biopolymers of high structural complexity, most of which are highly resistant to degradation [1]. However, in nature, there are microorganisms that are capable of modifying these structures thanks to the production of highly specific enzymes such as cellulases, xylanases, laccases, and peroxygenases [2]. These biomolecules are used as useful tools in various biotechnological processes due to their numerous technical and economic advantages [3].

Among the most well-known species of fungi for these purposes are Ascomycetes and Basidiomycetes. These microorganisms are the most abundant fungi in the upper layer of the soil. In addition to their well-known structural differences in hyphal size and mycelial structure, which make them very efficient in nutrient uptake and soil aggregation, basidiomycetes are generally considered more active in lignin degradation, while ascomycetes are mainly responsible for the degradation of plant cell wall polysaccharides (cellulose and hemicellulose and pectins). Indeed, both phyla contribute jointly or sequentially to the decomposition of plant residues [4].

Basidiomycetes, on the other hand, possess two types of extracellular enzymatic systems necessary to degrade plant biomass: (1) a hydrolytic system responsible for polysaccharide degradation, consisting mainly of xylanases and cellulases, and (2) a unique oxidative ligninolytic system, capable of degrading lignin and opening phenyl rings, comprising mainly laccases, ligninases, and peroxidases [5]. In contrast, ascomycetes have stronger cellulolytic activity, with the cellulase enzyme complex being prominent, composed mainly of endoglucanases, cellobiohydrolases, and β-glucosidases with an important role in soil ecosystems. Notable fungi such as *Aspergillus*, *Chaetomium,* and *Trichoderma* have been shown to improve straw decomposition in the field. Furthermore, fungi found in the rhizosphere of plants, such as *Aspergillus*, *Penicillium,* and *Trichoderma*, promote plant growth by producing enzymes, inorganic acids, and phosphatases, which help in the solubilization and mineralization of organic phosphorus. However, there is another great variety of fungi belonging to the Ascomycota division that present high degradative power and, therefore, may be potentially useful in the production of enzymes and in the degradation of lignocellulosic biomass, but they have not been widely studied. This is the case of the *Curvularia* genus, which, despite being considered an important phytopathogen that causes serious damage to a large number of crops, has a complex enzymatic battery with the expression of enzymes that modify lignin. Within this genus, *Curvularia lunata* is one of its most representative species [6].

However, there are many species that have not been studied in depth, such as *C. kusanoi*, so the objective of this research is to evaluate the enzymatic and microbiological characteristics of the isolated strain *C. kusanoi* L7.

## 2. Materials and Methods

*Microorganism*: *C. kusanoi* L7 strain, isolated from lemon tree, with the number of nucleotide sequences registered in the GenBank and accession number KY795957.

### 2.1. C. kusanoi L7 Growth on Different Culture Media

To evaluate the morphological characteristics of the fungus *C. kusanoi* L7, three culture media (Biolife, Mascia Brunelli, Milan, Italy), PDA (potato agar), CYE (Czapeck agar), and AMA (malt agar), were evaluated. From the pure culture of this strain on PDA, a 1mm fragment of mycelium was punctured in each plate and these were incubated in complete darkness at 25 and 30 °C. The diameter of the colonies was measured with a ruler at 3, 6, and 9 days of mycelia growth. The distinctive features of the conidial structures were determined at 9 days in water agar. Control groups without microorganisms were used for each type of culture.

### 2.2. Growth on 2% Cellulose Agar

For the qualitative evaluation of cellulolytic activity, the methodology proposed by Teather and Wood [7] was used. The strain was processed through streaking in petri dishes containing 2% microcrystalline cellulose and base agar as culture medium (Biolife, Mascia Brunelli, Italy). The cultures were incubated in complete darkness in an incubator for 7 days at 30 °C. The appearance of degradation halos around free colonies was taken as a positive test response; the extent of degradation was visually evaluated. The experiments were carried out in triplicate, using noninoculated medium as a negative control.

### 2.3. Growth on Tannic Acid

The *C. kusanoi* L7 strain was seeded in Nobles medium [8], with tannic acid as the sole carbon source, using 1 mm of the pure culture for 5 days of growth on PDA. The plates were incubated at room temperature in complete darkness for 7 days and the power index of the diffusion halos was evaluated as the relationship between the areas of fungal growth and degradation of tannic acid. The experiments were carried out in triplicate, using noninoculated medium as a negative control.

### 2.4. Determination of C. kusanoi L7 Carbon Mineralization of Raw Wheat Straw by Gas Spectroscopy—Total Attenuated Reflection Infrared Spectroscopy with Fourier Transform (ATR-FT-IR) Assisted

To evaluate the ability of the strain to degrade high-fiber substrates, the carbon mineralization of raw wheat straw was evaluated by gas spectroscopy. The fiber composition of this substrate is shown in the Appendix A. The experiment was carried out in solid fermentation, where 3 g of substrate was placed in 200 mL glass bottles closed with pierceable rubber stoppers, minimal salt medium was added (1 mL), and 1 cm^2^ of the mycelium grown in PDA of the *C. kusanoi* L7 strain was inoculated. Control groups without microorganisms were used for each culture. The production of CO_2_ was monitored through aliquots of the gas product of fermentation by puncturing the caps and injection into a gas chromatograph (Trace GC, Thermo Electron, Perkin Elmer, Waltham, MA, USA) equipped with a thermal conductivity detector. The degree of degradation of the raw wheat straw at the end of the test was evaluated by Total Attenuated Reflection Infrared Spectroscopy with Fourier Transform (ATR-FT-IR) in a Perkin Elmer equipment, Waltham, MA, USA with ATR diamond base and MCT/A detector. The scans were carried out from 4000 to 400 cm^−1^ [2].

### 2.5. Lignocellulolytic Capacity of the C. kusanoi L7 Strain in Solid Submerged Fermentation of Sugarcane Bagasse and in Wheat Bran

A 3 cm^2^ fragment of pure culture grown in PDA medium was used as inoculum in flasks containing 4 g of sugarcane bagasse and 100 mL of citrate buffer (50 mM, pH 5.0) and in flasks containing 3 g of Allbran-Kellogg’s cereal based on wheat bran (the composition of these substrates is shown in Appendix A) and 100 mL of citrate buffer (50 mM, pH 5.0). Control groups without microorganisms were used for each culture. The flasks were incubated on an orbital shaker at 120 rpm for a period of 10 days at 30 °C. Fermentation samples were taken every 24 h, the content of each flask was filtered through a Büchner funnel, and the resulting liquid was centrifuged (4 °C, 10,000 rpm, 3 min) [9].

Cellulase, laccase, and lignin peroxidase activity were determined in the supernatants from the fermentation (enzymatic extract). A total of 100 μL of enzymatic extract was used in each enzyme activity assay. A final reaction volume of 1 mL was used for these determinations in each method. Endo-1,4-β-glucanase (CMCase) was determined on carboxymethylcellulose 2.2% (*w/v*) in 50 mM sodium citrate buffer, pH 5. The exo-1,4-β-glucanase (PFase) activity was quantified on Whatman No. 1 filter paper (50 mg) in 0.6 M sodium acetate buffer, pH 6.0. All reactions were incubated at 50 °C for 30 min and the content of reducing sugars released was determined by 3,5-dinitrosalicylic acid (DNS) method [10], and enzyme activities were expressed as micromoles of glucose released per minute [11]. Laccase activity was determined by degradation of syringaldazine. The reaction mixture formed by 100 μL of syringaldazine (50 mM in ethanol) and 800 μL of citrate buffer (50 mM, pH 4.5) was incubated at 30 °C for 1 min. The syringaldazine oxidation reaction was monitored kinetically for 1 min under aerobic conditions at 530 nm. One unit of laccase activity (U) was considered as the amount of enzyme that catalyzes the conversion of 1.0 µmol of syringaldazine per minute [12]. The lignin peroxidase activity was determined by the H_2_O_2_-dependent oxidative dimerization of 2,4-dichlorophenol at 20 °C. A unit of enzyme activity was considered as the amount of enzyme that can increase 1.0 unit of absorbance per minute [13].

### 2.6. Statistical Analysis

For the *C. kusanoi* L7 carbon mineralization of raw wheat straw by gas spectroscopy–Total Attenuated Reflection Infrared Spectroscopy with Fourier Transform (ATR-FT-IR) assisted, a completely randomized design was used. The measurement was performed at three-day intervals for one month. Arithmetic means and standard deviation were calculated and the differences between the means were established according to Duncan [14] with the help of the InfoStat v1 statistical program [15].

For the evaluation of the lignocellulolytic capacity of the *C. kusanoi* L7 strain in solid submerged fermentation of sugarcane bagasse and in wheat bran, each enzyme analysis was processed according to simple variance analysis in order to evaluate the effect of time on their production, with the help of the InfoStat v1 statistical package [15]. Duncan’s test [14] was used when necessary to discriminate differences between the means.

## 3. Results

### 3.1. C. kusanoi L7 Growth on Different Culture Media

The macroscopic characteristics in different culture media of the fungus *C. kusanoi* L7 are shown in Figure 1, and the influence of temperature on fungal growth in different culture media is included as Appendix A. The greatest mycelial growth was found in CYE agar, although the strain also grows rapidly in AMA, but not in PDA medium, where it presented less growth. Particular differences were observed in terms of the morphology of the strain depending on the culture medium evaluated. In general, the colonies presented velvety to woolly surfaces, with young white mycelium becoming different over time in terms of color depending on the medium.

The results can be related to the different composition of the media used in this study. In the case of CYE (a medium that is used mainly for the maintenance of strains and taxonomic studies), sucrose (~30 g/L) is found as the main sugar. In the case of AMA (medium indicated for the growth of fungi and yeasts), glucose is the predominant sugar (~20 g/L). Lastly, in the case of PDA (a medium that stimulates mycelium sporulation for isolation and identification studies), dextrose (~20 g/L) is found as the main sugar. The macroscopic characteristics of fungal cultures also depend on the carbon–nitrogen (C/N) relationship, which is different in each culture medium evaluated [16].

It is known that the microorganism can present a different adaptation to different nutritional conditions, directly affecting its development [17]. Elements such as carbon, nitrogen, sulfur, iron, and other minerals are essential for the growth and enzymatic production of fungi, where carbon is the most important nutrient. Compounds such as ammonium chloride, peptone, and malt extract are used as nutritional nitrogen supply [17]. According to studies by Torres et al. [18] the carbon–nitrogen relationship is another determining factor that affects the formation of the mycelium and the fruiting body of higher fungi. In the present study, these three media have a very different carbon–nitrogen ratio, an aspect that supports the differences found: in the PDA medium there is no carbon–nitrogen ratio, CYE has a ratio of 10, while malt agar has a ratio of 1. The difference in carbon–nitrogen ratio between the media used is due to their composition, where the PDA medium only provides complex carbohydrates and mineral salts necessary for the development of the fungi and, in the case of CYE, in addition to sucrose, sodium nitrate is provided and, finally, in the medium with malt agar, nitrogen is provided by peptone.

On the other hand, temperature has a significant effect on the development of microorganisms [19]. In the specific case of fungi, an optimal growth temperature of 30 °C is reported for the vast majority of them. In the present investigation, greater mycelial development was evidenced at 30 °C than at 25 °C in all the culture media evaluated (presented in Appendix A). Regarding the microscopic characteristics of the cultures, it is necessary to point out that the fruiting bodies were not visualized in any of the evaluated media. The distinctive features of the conidial structures were determined by sowing the microorganism in a simpler culture medium (water agar) (Figure 2), which is reported in this study as a new characteristic for the strain under evaluation.

Conidia formation is a response to unfavorable growing conditions, such as nutrient deficiencies or nutritional depletion. The latter is an extreme condition for vegetative growth; however, nutrient-poor media with a low carbon and nitrogen source favor the formation of chlamydospores with suppression of vegetative growth as a function of nutritional stress [20]. For these reasons, the fruiting bodies could be visualized in water agar since this medium has a minimum amount of nutrients.

The fruiting bodies observed presented the distinctive characteristics of the *Curvularia* genus, which is characterized by curved, spindle-shaped, ovoid, or ellipsoidal fragmoconidia, brown in color with a typically wider central cell that is darker in color, as well as simple conidiophores and branched in some cases [21]. Similar conidial characteristics can also occur in other genders, such as Bipolaris. According to Piontelli [21], *Cochliobolus*, *Bipolaris,* and *Curvularia* make up a complex of taxonomically confusing species. Due to the constant changes in the nomenclature of some of their asexual members (*Bipolaris* and *Curvularia*), which differ mainly based on the morphology of their conidia, this is a situation that is sometimes very difficult due to the fact that, in both genera, some species have similar conidial characteristics.

Maciel et al. (2006) [22] evaluated the characteristics of the fungi *Duddingtonia flagrans* (CG768), *Actinella robusta* (I31), and *Monacrosporium thaumasium* (NF34A) in different commercial culture media. The observation of the conidial structures characteristic of each strain, as in the present study, were observed using water agar.

### 3.2. Growth of C. kusanoi L7 in Medium with 2% Cellulose and on Tannic Acid 

*C. kusanoi* is a microorganism that has been poorly studied; for these reasons, it was decided to analyze its morphological and growth characteristics in various culture media, as well as its degradative potential. For this purpose, two semi-quantitative methods were used, such as growth in 2% cellulose (method used to evaluate cellulolytic capacity) and growth in tannic acid (method to evaluate possible ligninolytic activity). Both determinations allow the analysis of the possible expression of the main enzymes that modify the plant cell wall. The results of both studies are shown below.

In both methods (growth in 2% cellulose and growth in tannic acid), Congo red is incorporated as a dye in the preparation of the culture medium so that, if the fungus has cellulolytic or ligninolitic capacity, degradation halos or diffusion zones are evident, where, in addition to being indicative of this type of activity, it also serves as a method of comparison between microorganisms when measuring the power index, which relates the diameter of the hydrolysis halo with the diameter of the colony.

*C. kusanoi* L7 strain in a medium with 2% cellulose developed rapid growth and notable degradation halos (10 ± 5 mm). According to Rosyidaa et al. [23], the incidence of species capable of using native cellulose is high when starting from soil isolates, as this ecosystem constitutes a promising source of cellulolytic organisms. However, isolates from plant material (as in this case) allow us to obtain specificity of action of fungi on the fiber substrate [24]. In this way, the natural habitat conditions of the species are maintained and obtaining strains with greater degradative power is guaranteed.

Regarding the growth in tannic acid as a derivative of higher molecular weight of lignin, it is a very useful test to evaluate if the microorganism is capable of producing ligninolytic enzymes. In the present study, it was possible to detect the presence of diffusion zones or dark halos around the colony for a potency index of 4. This result is in agreement with the potency indexes reported for basidiomycetes fungi that have a unique oxidative ligninolytic system [5]. That is why this strain, despite being an ascomycete fungus characterized by a stronger cellulolytic activity [6], is also capable of expressing oxidative enzymes. The secretion capacity of oxidase enzymes is only present in a small number of microorganisms given the high structural complexity of lignin [25]. For these reasons, the growth and appearance of degradation halos in a medium where tannic acid (high-molecular-weight derivative of lignin) is the only carbon source can be associated with the ability of the strain to express specific enzymes that modify phenolic compounds with a structure similar to lignin.

The results of the semi-quantitative tests justify the following study of mineralization of a more complex substrate such as raw wheat straw, where the microorganism needs the expression of its enzymatic system for effective degradation of the fibrous material.

### 3.3. Determination of C. kusanoi L7 Carbon Mineralization of Raw Wheat Straw by Gas Spectroscopy–ATR-FT-IR Assisted 

The results of the mineralization study of raw wheat straw (highly complex fibrous substrate resistant to degradation) by the action of the lignocellulolytic fungus *C. kusanoi* L7 are shown in Figure 3. 

From the 10th day of fermentation, a drastic decrease in CO_2_ production was observed, which may be related to the decrease in nutrients like cellulose and hemicellulose and the beginning of the degradation of the most complex components of the plant cell wall. 

However, from day 20 of fermentation, the mineralization levels of the substrate begin to increase again at the expense of the degradation of more complex compounds, such as lignin. In the ATR-FT-IR study (Figure 4), it can be observed that there is not only degradation of structures belonging to the carbon skeleton of cellulose fibers (signals 1–4) but also of structures associated with the carbon skeleton of lignin (signals 5–7); therefore, the decrease in the intensity of these bands indicates their structural modification. In the present study, the decrease in the intensity of these signals corroborates the previous approaches that suggest the expression of enzymes that modify lignin and allow a more exhaustive degradation of the plant wall in the treatments with *C. kusanoi* L7.

According to the scientific literature, the Curvularia genus stands out among ascomycete fungi for presenting a great degradative power, especially of complex compounds. However, the species under study has not been investigated in this regard, which makes it difficult to compare results. However, studies can be mentioned where other species are used, such as *Curvularia lunata* [26] or *Curvularia senegalensis* [27], where the degradation products are also analyzed using infrared spectroscopy. These studies show the powerful degradative activity of these fungi when degrading complex substances such as polyethylene and polyurethane.

### 3.4. Lignocellulolytic Capacity of the C. kusanoi L7 Strain in Solid Submerged Fermentation of Sugarcane Bagasse and in Wheat Bran

Thus, the results of carbon mineralization and the evidence of the degradation of cell wall components carried out in the present investigation allow us to affirm that the fungus *C. kusanoi* L7 is also capable of expressing specific enzymes responsible for breaking down fiber components that are difficult to degrade. For these reasons, it was decided to quantitatively evaluate the enzymatic activity of the main enzymes present in fiber degradation, such as the cellulase complex and oxidative enzymes such as laccases and peoxidase. Sugarcane bagasse and raw wheat straw were used for this study as complex fibrous substrates, which are currently considered one of the main wastes of the tropical agricultural industry.

The determination of the enzymatic activity of *C. kusanoi* L7 in both types of substrates are shown in Table 1, Table 2, Table 3 and Table 4. Table 1 and Table 2 show the cellulolytic capacity of the strain when producing the main enzymes of the cellulase complex (endo-1,4-β-glucanase and exo-1,4-β-glucanase). Another important aspect was the higher enzyme activity values obtained during the first 24 to 72 h of fermentation, a fundamental aspect for fermentation processes of candidate strains.

As can be seen in the table, the highest value of endoglucanase activity was obtained at 24 h of fermentation and this begins to gradually decrease after 96 h. This type of activity begins by randomly breaking the b-glycosidic bonds within the cellulose molecules, producing new attack sites complemented by exoglucanases. As a result, there is a rapid decrease in the chain length and a slow increase in the reducing groups.

Regarding the activity of exoglucanases, the highest values are obtained after 48 h of fermentation. These enzymes gradually attack the cellulose molecules at the nonreducing ends, releasing cellobiose subunits. They are active against crystalline cellulose and present a highly co-operative synergistic action in the presence of endoglucanases. The latter attack the amorphous regions of the cellulose fibers, thus creating new sites for the exoglucanases to continue the degradation of the fiber.

As can be seen in Table 2, in wheat bran, the highest value of endoglucanase activity is reached at 72 h of fermentation, where it begins to gradually decrease. Regarding exoglucanase activity, the highest values are obtained at 24 h of fermentation and it gradually decreases during the rest of the fermentation.

Regarding laccase and peroxidase activity in sugarcane bagasse, it can be noted that the highest values of enzymatic activity are reached after 168 h (7 days) but not for the production of peroxidase, where its highest activity is obtained after 96 h of fermentation. The latter was only detected in the fermentation supernatants at 96, 120, and 144 h; neither before nor after was peroxidase activity detected.

Regarding laccase and peroxidase activity in wheat bran, similar results were obtained to those of sugarcane bagasse fermentation, where the highest values of laccase enzymatic activity are reached after 168 h and peroxidase production is detected in the fermentation supernatants only at 96, 120, and 144 h, with the highest peroxidase activity also occurring after 96 h of fermentation.

The ligninolitic activity in both substrates (Table 3 and Table 4) showed that, besides having the same kinetics where the maximum potential for laccase production is reached at seven days of fermentation and at the fourth day for peroxidase, the values are higher when the microorganism is degrading wheat bran.

## 4. Discussion

As shown in the results, *C. kusanoi* has the ability to grow in different media. In addition, its characteristic production of enzymes allows it to degrade both the cellulose and the lignin fraction. The study of carbon mineralization assisted by infrared spectroscopy allows us to demonstrate how, through its fermentation, this microorganism is able to grow at the expense of the degradation of both simple compounds and compounds of greater structural complexity.

The biodegradation process of lignocellulosic substrates is a difficult phenomenon and its difficulty is closely related to the content of cellulose, hemicellulose, and lignin present in the material. For these reasons, it can be stated that the CO_2_ production of a microorganism is affected by the quality of the organic material that is degraded. Raw wheat straw, according to the Spanish Foundation for the Development of Animal Nutrition [28], is a substrate that presents, on average, 72% neutral detergent fiber (NDF), which is distributed in 38% cellulose, 25% hemicellulose, and 10% lignin. In the present case, the composition of raw wheat straw (study presented in Appendix A) indicates similar values in the fiber components (67.54% neutral fiber, 28.02% cellulose, 25.51% hemicellulose, and 7.13% lignin). The fibrous characteristics of this type of substrate result in the microorganism needing the expression of certain enzymes that catalyze the degradation of these components, resulting in a slower degradation rate.

Regarding lignocellulolytic capacity of the *C. kusanoi* L7 strain in solid submerged fermentation of sugarcane bagasse and in wheat bran, it is important to note that the strain presented higher cellulolytic activity in sugarcane bagasse. It is known that, in the presence of substrates that are more complex and with higher fiber content as in sugarcane bagasse, the induction of the cellulolytic system occurs with the consequent increase in the production of cellulases [29]. From a structural point of view, sugarcane bagasse has a greater amount of the components of the acid fraction of the fiber, highlighting a greater amount of cellulose and lignin (as presented in Appendix A) than in wheat bran. However, although sugarcane bagasse presents greater structural complexity than wheat bran [28], the production of these enzymes had a similar behavior when the highest cellulolytic enzymatic activity was reached during the first 24 to 48 h.

Candidate strains for the development of enzymatic production fermentation processes are those capable of expressing their maximum capacity during the first hours of fermentation, key aspects to reduce working time and optimize the technological process [30]. However, although the cellulase production of this strain can be considered high, it is below the values reported for *T. viride* (M5-2) in sugarcane bagasse, which presented high cellulolytic activity after 24 h and reached its maximum expression at 72 h for exoglucanase (1.84 IU/g of MS) and endoglucanase (7.26 IU/g of MS) [31]. Likewise, it is necessary to highlight the characteristics of this type of enzymes, since, in addition to being produced early in the fermentation process, they also present high thermal stability and enzymatic activity in a wide pH range (6–10), which makes them important tools for the biotechnology industry [30].

In a previous study, where the activity of different fungal isolates was evaluated, the strain *C. kusanoi* L7 was evaluated on grass hay, finding a similar expression of cellulolytic enzymes to those found in the present study in sugarcane bagasse. Grass hay has a similar composition in terms of fiber to bagasse, with acid detergent fiber values close to 70%. On the other hand, the characteristics of the purified laccases of this fungus were also evaluated, finding that they are active in alkaline pH, which gives them high stability. Likewise, the effect of temperature on enzymatic activity was evaluated, finding an optimal activity range between 30 and 40 °C. *C. kusanoi* L7 laccases are thermostable and maintain residual enzymatic activity at temperatures between 60 and 70 °C [32]. 

Comparatively, in the present study, the enzymatic crude obtained from the fermentation presented a higher laccase activity (1.64 IU/mL). This is a highly significant aspect since, despite the fact that this microorganism is an ascomycete fungus, its laccase productions are in the range of those produced by several species of basidiomycete fungi defined as high producers, for example, *Tramentes versicolor* in corn silage with a laccase activity of 0.18 IU/mL [33], *Pleurotus ostreatus* in oil palm trunk with 0.218 IU/mL [34], *Coriolopsis gallica* in sawdust waste with 4.8 IU/mL [35], and *Pleurotus pulmonaris* in bagasse with 2.1 IU/mL [36].

Regarding the production of ligninolytic enzymes in sugarcane bagasse (Table 3), peroxidase activity is lower than laccase activity, which reached its maximum activity at 120 h of fermentation. It is important to point out that the volumetric activity of the laccases that are expressed in this medium is around 30 times lower than the activity of the laccases that were expressed in the solid submerged fermentation of wheat bran. This aspect may be due to the characteristics of these culture substrates, since laccases are induced in a variable way according to the effect of different factors, such as the concentration of different sugars and aromatic compounds [37]. Although these compounds have not been evaluated in the substrates used, it is known that many of these compounds (sugars and aromatic compounds) are found in wheat bran, so this substrate is used as a preferred medium for the induction of laccase enzymes in different fungi, such as microorganisms such as *Pseudolagarobasidium acaciicola*, *Pleurotus ostreatus*, *Trametes versicolor*, *Pleurotus pulmonarius*, and *Pycnoporus sanguineus* [38]. On the other hand, it is known that the expression of cellulases and hemicellulases is repressed in the presence of D-glucose, while the opposite occurs for laccases, where abundant hydroxycinnamic acids present in wheat bran, particularly p-coumaric and ferulic acids, stimulate the production of laccases as they constitute precursors of lignin synthesis according to Neifar et al. 2009 [37]. These authors also studied the effect of copper sulfate on the induction of laccases by the white-rot fungus *F. fomentarius* in solid-state fermentation of wheat bran. In the study, a laccase activity close to 6.4 IU/mL was reached after 13 days of incubation. When the wheat bran medium was supplemented with 2 mmol l(-1) copper sulfate, laccase activity increased by threefold in comparison to control cultures, reaching 27.8 IU/mL. This study showed, in addition to the effect of the substrate, the usefulness of the enzyme induction by copper sulfate.

Regarding the expression of laccases in the genus *Curvularia*, there are few reports in the literature; however, some authors [39] found values of laccase activity in *Cochliobolus* (the teleomorphic state of different types of *Curvularia*) similar to those presented in this study. These results are a measure of the high potential that this genus can have in the degradation of recalcitrant compounds such as lignin. On the other hand, it is evidenced that the laccase enzyme production pattern of this strain is very similar to most of the lignolytic strains reported as oxidase-producing, where the highest laccase activity is reached 7 days after fermentation [40]. The lignin enzymatic degradation process could increase the number of pores and the available surface area. This process allows better access of the xylanase and cellulase enzymes, in addition to directly improving the total hydrolysis yields [41]. Despite the fact that biological methods are capable of reducing the lignin content of plant biomass [40], the efficiency of this process is related to both the enzymatic production of the fungus and the specific treatment time of the substrate, which can vary from 7 to 84 days [42].

*C. kusanoi* L7, despite being an ascomycete fungus (characterized by being fundamentally cellulolytic, therefore, with greater specificity for cellulose), has the advantage of also producing high concentrations of laccase enzymes compared to the capacity of basidiomycete strains. Therefore, it not only presents specificity towards cellulose but, at the expense of its oxidase enzymes, it can also efficiently degrade lignin, which makes it a useful microorganism from a biotechnological point of view, since a more exhaustive degradation of the cell wall can be achieved by combining both types of enzymatic activity on the plant cell wall. In general, the lignocellulolytic capacity of this microorganism is sufficient to assess its use in fiber pretreatment processes, and its high production of laccases makes it interesting as a strain with potential for obtaining this type of enzyme with great industrial application.

## 5. Conclusions

The characteristics of the *C. kusanoi* L7 strain, its ability to grow in different culture media, efficiently mineralize carbon, and express a versatile enzymatic battery in complex high-fiber substrates, make it an ascomycete with great lignocellulolytic potentialities to evaluate in processes of bioconversion of plant biomass.

## Figures and Tables

**Figure 1 jof-10-00807-f001:**
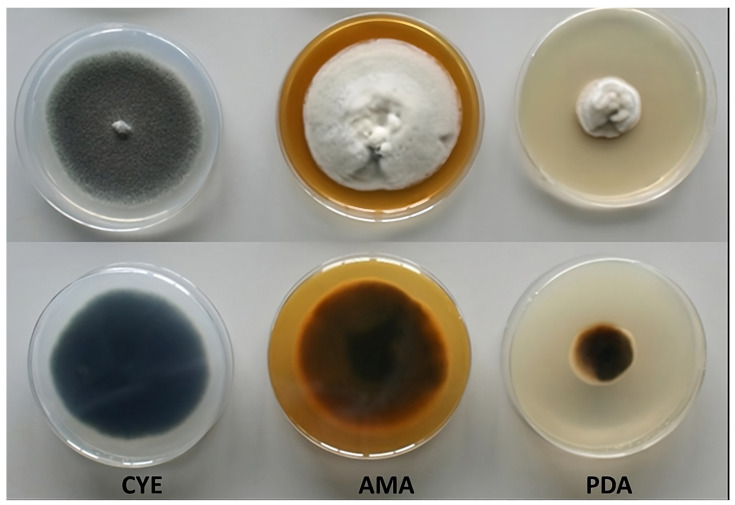
Growth of the fungus *C. kusanoi* L7 in different culture media (7 days, 30 °C, pH 5.4, complete darkness, and 85% of relative humidity). CYE (Czapeck agar), AMA (malt agar), and PDA (potato agar). Top Panel: front of plate; Bottom Panel: back of plate.

**Figure 2 jof-10-00807-f002:**
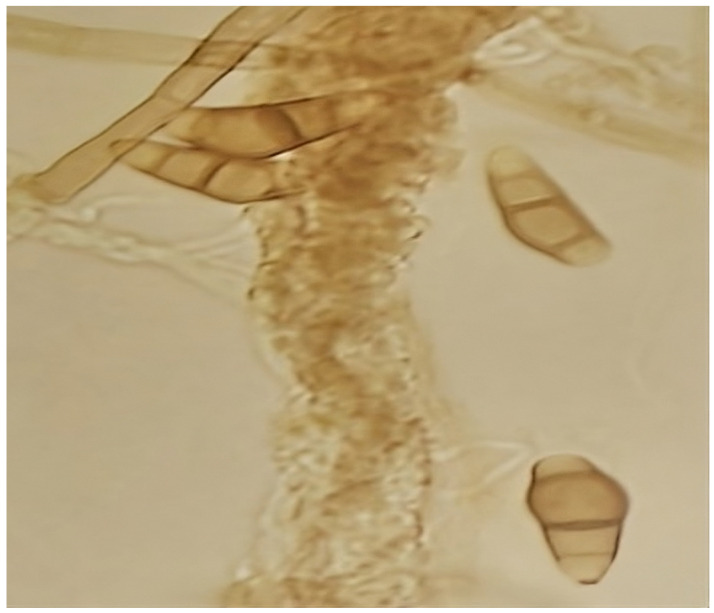
Microscopic characteristics of *C. kusanoi* L7 fruiting bodies in water aga; 10-day-old colonies.

**Figure 3 jof-10-00807-f003:**
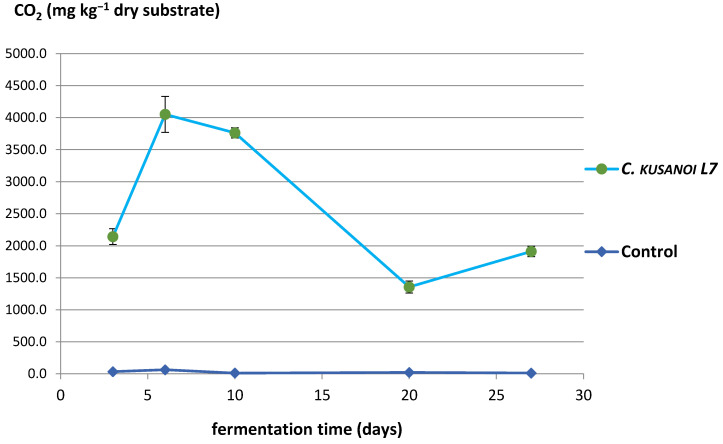
*C. kusanoi* L7 carbon mineralization of raw wheat straw.

**Figure 4 jof-10-00807-f004:**
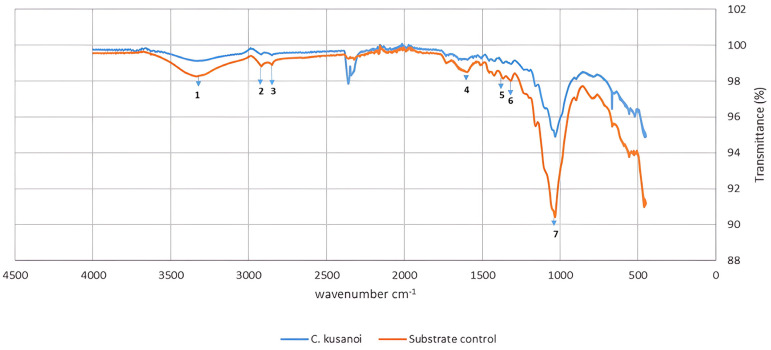
ATR−FT−IR spectrum of carbon mineralization of raw wheat straw by *C. kusanoi* L7. Substrate control: wheat straw. Study range from 4000 to 400 cm^−1^. ^1–7^ correspond to the main groups affected by the fungal degradation. ^1^ Band at 3400 to 3200 cm^−1^: stretching vibration of O-H groups phenolic and aliphatic. ^2^ Band at 2930 cm^−1^: CH_3_ and CH_2_ stretching vibration of C-H bonds. ^3^ Band at 2850 cm^−1^: OCH_3_ groups vibration. ^4^ Band at 1610 cm^−1^: aromatic C=C double bonds. ^5^ Band at 1420 cm^−1^: vibrations of lignin phenylpropane aromatic skeleton. ^6^ Band at 1333 cm^−1^: aliphatic C-H bonds vibrations (CH or CH_2_ groups). ^7^ Band at 1035 cm^−1^: O-CH_3_ bonds vibrations of guaiacyl and syringyl type units.

**Table 1 jof-10-00807-t001:** Cellulolytic activity (endo-1,4-β-glucanase and exo-1,4-β-glucanase) of the *C. kusanoi* L7 strain in sugarcane bagasse.

Cellulolytic Activity (IU/mL)	Fermentation Time (Hours)							
	24	48	72	96	120	144	168	SE and *p*
**Endo-1,4-β-glucanase**	**2.87 ^g^**	2.11 ^e^	2.41 ^f^	1.90 ^d^	1.28 ^c^	1.13 ^b^	0.977 ^a^	±0.22 *p* = 0.0004
**Exo-1,4-β-glucanase**	0.35 ^b^	**0.90 ^d^**	0.61 ^c^	0.018 ^a^	0.008 ^a^	0.003 ^a^	0.0006 ^a^	±0.36 *p* = 0.0021

^a–g^ Means with different letters in each row differ at *p* < 0.05 (Duncan, 1955). SE: standard error, *p*: level of significance. The values highlighted in bold correspond to the highest enzymatic activity detected in the analysis.

**Table 2 jof-10-00807-t002:** Cellulolytic activity (endo-1,4-β-glucanase and exo-1,4-β-glucanase) of the strain *C. kusanoi* L7 in submerged solid medium of wheat bran.

Cellulolytic Activity (IU/mL)	Fermentation Time (Hours)							
	24	48	72	96	120	144	168	SE and *p*
**Endo-1,4-β-glucanase**	0.608 ^e^	0.402 ^d^	**0.921 ^f^**	0.185 ^c^	0.098 ^b^	0.035 ^a^	0.029 ^a^	±0.022 *p* = 0.0003
**Exo-1,4-β-glucanase**	**0.55 ^c^**	0.24 ^b^	0.20 ^b^	0.08 ^a^	0.07 ^a^	0.07 ^a^	0.03 ^a^	±0.011 *p* = 0.045

^a–f^ Means with different letters in each row differ at *p* < 0.05 (Duncan, 1955). SE: standard error, *p*: level of significance. The values highlighted in bold correspond to the highest enzymatic activity detected in the analysis.

**Table 3 jof-10-00807-t003:** Ligninolytic activity (laccase and peroxidase) of the strain *C. kusanoi* L7 in sugarcane bagasse.

Ligninolytic Activity (IU/mL)	Fermentation Time (Hours)							
	48	72	96	120	144	168	192	SE and *p*
**Laccase**	-	0.13 ^a^	0.30 ^b^	0.38 ^b^	0.51 ^d^	**0.84 ^e^**	0.49 ^c^	±0.61 *p* = 0.0012
**Peroxidase**	-	-	**0.21 ^c^**	0.11 ^b^	0.002 ^a^	-	-	±0.36 *p* = 0.043

^a–e^ Means with different letters in each row differ at *p* < 0.05 (Duncan, 1955). -: no activity found, SE: standard error, *p*: level of significance. The values highlighted in bold correspond to the highest enzymatic activity detected in the analysis.

**Table 4 jof-10-00807-t004:** Ligninolytic activity (laccase and peroxidase) of the strain *C. kusanoi* L7 in submerged solid medium of wheat bran.

Ligninolytic Activity (IU/mL)	Fermentation Time (Hours)							
	48	72	96	120	144	168	192	SE and *p*
**Laccase**	-	0.24 ^a^	0.46 ^b^	0.53 ^c^	0.81 ^d^	**1.64 ^e^**	0.6 ^c^	±0.44 *p* = 0.0023
**Peroxidase**	-	-	**0.18 ^c^**	0.06 ^b^	0.002 ^a^	-	-	±0.36 *p* = 0.0034

^a–e^ Means with different letters in each row differ at *p* < 0.05 (Duncan, 1955). -: no activity found, SE: standard error, *p*: level of significance. The values highlighted in bold correspond to the highest enzymatic activity detected in the analysis.

## Data Availability

The microorganism sequences were deposited to GenBank under accession number KY795957.

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
