# Peer review of "Microbiological Aspects and Enzymatic Characterization of Curvularia kusanoi L7: Ascomycete with Great Biomass Degradation Potentialities"

_jof, 2024, doi:10.3390/jof10120807_

Round 1
Reviewer 1 Report
This study aims to evaluate the microbiological and enzymatic characteristics of the fungal strain Curvularia kusanoi L7. The research is of significant importance for finding new microbial strains that can effectively degrade fibers. The experimental design is rational and supported by sufficient data. However, there are still some issues that need to be addressed and further improvements made.
(1) The clarity and aesthetics of the figures and tables in this paper are both low and need improvement.
(2) On Page 5, Figure 2, the two diagrams illustrate the same issue; it is suggested to select only one to avoid data repetition.
(3) On Page 5, lines 176-177, the C. kusanoi L7 strain shows rapid growth and forms a significant degradation halo in a medium containing 2% cellulose. Please provide the image and explain why the cellulose content is 2%.
(4) The logical flow of the article is not strong, and there is a lack of connectivity between the media used in sections 3.1 and 3.2 of the results and discussion.
(5) This study lacks a control group without microbes, and the corresponding physicochemical indicators have not been analyzed or explained.
(6) On Page 4, lines 146-147: "In the present investigation, a greater mycelial development was evidenced at 30ºC than at 25ºC in all the culture media evaluated." Humidity, light, and pH all affect fungal growth; please provide specific data.
(7) Section 3.3 only describes based on the figures and does not combine previous research for reasonable speculation and analysis.
(8) Section 3.4 has limited results; please provide annotations for the four tables and a rational analysis.
(9) On Page 9, lines 262-264, how were the data showing the highest enzyme activity within 24-48 hours obtained? Please explain.
(10) The conclusions section mentions the biotransformation process, but the paper does not address the process or mechanism of biotransformation.
(11) There are few references and the cited literature is not recent enough; the format also needs further improvement.
Reviewer 2 Report
The work of Vazquez et al., entitled "Microbiological aspects and enzymatic characterization of Curvularia kusanoi L7: Ascomycete with great biomass degradation 2 potentialities" investigates the growth of Curvularia kusanoi L7 on different substrates and explores its lognocellulolytic potential. Indeed there is limited research on this specific microorganism and Authors attempt to demonstrate some unique features regarding the enzymatic toolbox of the microorganism. However, there are some major issues and inquiries regarding the experimental procedures and discussion.
L39: What do the authors mean by "best results"? Regarding what aspect?
L40: "there is another variety of fungi". Are you referring to a particular group? Literature is missing.
L86: to which experiments are the Authors referring?
L93: What does a "rate of 3 cm2" mean? a rate is expressed per time units.
L94-95: Where these paterials commercial? How were they supplied? Why were they ised in different quantities? This could significantly affect the enzymes quantity expressed by the microorganism.
LL106: Was the stock solution of syringaldazine prepared in ethanol at a concentration of 5 mM?
L107: The Units for every enzymatic activity should be defined.
L112: the fact that there are 2 "Statistical analysis" paragraphs is confusing. They should be combined in one and edited accordingly.
L121: Are there any numerical data, since the groth was meanured overtime as stated in Materials and Methods?
L125-130: How does the medium causes differentiation in morphology? Is this expected? Is this common among similar microorganisms?
L136: Indicate Panel A and B on the figure.
L137: To which certain fungal species are the Authors referring?
L138-143: This is interesting information. However, how do these factors differs among the media that were used in this work? The discussion should be further oriented towards this dirrection in order to understand the differenced in the morphology on the microorganism.
L149-151: Perhaps this info could be discussed in more detail? What is the significance? Has this ever been noticed before for similar microorganisms?
Figure 2: Panels A & B should be indicated in the figure. What is the difference between them? Describe shortly what the reader can see there.
L156-160: Some rephrasing is required here for better clarity. What is it meant by "adverse condition of vegetative growth" and "their contribution of nutrients is zero"?
L171-174: The sentence should be edited, it is too complex and long. More details should be given on the "similar results" obtained from the literature.
L179-181: The substrate referred in the text is lignin? Do the soil microorganisms not have to face lignin? How the conclusion is reached that the plant biomass which ends up in soil is more rich in cellulose than the plant itself?
L184-185: What do the diffussion zones represent? And what is the significance of a "potency index of 4"?
L185-187: Please elaborate more on this background. Based on the loiterature what is the capasity of Ascomycete fungus to express lignin degrading microorganism? This could highlight the importance of your work.
L188: A more appropriate term would be "oxidative enzymes"
L190-191: "modify phenolic compounds with a structure similar to lignin". how is this concluded? this is rather an assumption, not a conclusion. the fact is that the microorganism accumulated tanic acid, not lignin. if lignin could be incorporated in the media and study its degradation then this would be a safer method to reach this conclusion.
Figure 3: In the Y-axis, the CO2 is expressed in mg per kg of what?
L202-203: Again this is an assumption. Hemicellulose is another candidate for carbon source that could be degraded by the microorganism.
L222-230: There is no need to present the legends of the tables in the main text. It would be more meaning full to highlight the most important results of the tables at this point.
Tables 1-4: Why all of the values of the row have the same SE and Signf? Also define what SE and Signf stand for. Ammend the "UI" units to "IU".
L246: The statement "its versatile enzymatic production is significant" is rather extravagant. Only four enzymatic activities were examined (and there is some uncertainty regarding their distinguishment).
L247-248: Again the sentence is rather misleading as no kinetics models were demonstrated in this work and no constants were calculated.
L254: What does NDF stand for?
L262-263: How sugarcane bagasse demosntrates greater structural complexity that wheat bran?
L272-273: Support this statement with data from literature and not general statements.
L277: What is a "volumetric activity"?
L279-281: Please elaborate more on the various ways on induction of laccases and how this information is valuable to interpret the results of this work.
L281-283: why is this information relevant to the results of this work? Discuss a little bit further this thought.
L285: Discuss further on this similarity, in order to evaluate it. What substrates were used, and what induction conditions? what was the final result obtained?
L295: Again the term "kinetics" in not the appropriate one. There is no kinetic model, just a calculation of the enzyme activity per day, demonstrating a trend.
Reviewer 3 Report
This paper reviews enzymatic characteristics of an ascomycete, Curvularia kusanoi L7. The subject of this paper is fundamental information for evaluating the usefulness and utilization of strains, so it can be said to be necessary information for readers. However, some major revisions are necessary for publication.
1. (Introduction) The introduction should specifically explain the general research background, trends in related research, the necessity of the research (issues raised), and the purpose of this study. Even if there is insufficient information on the species used in the study, sufficient introduction of trends that can be referenced can be provided, and the justification for using this strain should also be presented in the introduction.
2. (Results) The morphological characteristics and changes in growth in response to temperature and nutrients in the medium have been reported in many other published papers. Therefore, it is necessary to use appropriate references and summarize only the necessary information.
- Figure 1 requires a front and back indication of each characteristics.
Round 2
Reviewer 1 Report
After careful consideration of the manuscript, I regret to inform you that I cannot recommend it for publication in its current form.
My primary concerns are as follows:
Incomplete Assessment of Enzymatic Activity and Substrate Composition: The authors have focused on carbohydrate hydrolase and oxidoreductase in the Curvularia kusanoi L7 strain. While this provides some insights into the strain's potential for biomass degradation, the manuscript lacks a thorough evaluation of the chemical composition of the fermentation materials, specifically cellulose, hemicellulose, and lignin content. This information is crucial for assessing the strain's applicability in treating agricultural residues like straw. The authors should include these analyses to provide a more comprehensive understanding of the strain's degradation capabilities.
Furthermore, the study does not provide evidence of the strain's selectivity for lignin degradation over cellulose, which is a significant gap in the research. The emphasis on fungi with selective degradation capabilities is particularly relevant in industrial applications, and this omission diminishes the manuscript's impact.
Quality of Graphical Representations: The quality of figures and tables in the manuscript is inadequate. They are both low in clarity and lack aesthetic appeal. High-quality, clear, and informative figures are essential for effective scientific communication. I recommend that the authors seek the assistance of a professional graphic designer or utilize advanced software to enhance the resolution and layout of their figures.
Author Response
Dear reviewer, I am attaching the detailed responses to your review of the manuscript in a Word document below.
The new version of the manuscript with the corrections, as well as supplementary material and images, also edited and improved, have been uploaded to the platform again.
Reviewer 2 Report
I would like to thank the Authors for taking into consideration all of my comments.
The revised manuscript has been significantly improved and its contribution in the current existing knowledge has been highlighted.
There are a few minor comments that could be further addressed.
L92-93: What is the final volume of the culture?
L112-114: What volume of the supernatant was used for the reaction mixture? What was the final volume of the reaction mixture?
L166: How the Carbon-Nitrogen ration differs among the media?
L200-206: It would be more appropriate to include this paragraph under Section 3.2.
L224: Ammend "potency" to "power"
L271-278: It would be more appropriate to include this paragraph under Section 3.4.
L289: "at 24 hours" instead "after 24 hours"
L297-299: Please rephrase the sentences. In the previous sentence it is stated that "They (exoglucanases) are not very active against crystalline cellulose" while later it is stated that exoglucanases "would then proceed towards the crystalline regions of the fiber", which is contradictory.
Tables 2-4: Please amend the title of the "SE and P" column as in Table 1. However, it is still not understood what does this standard error stand for. Usually, a standard deviation expresses the deviation of a mean value from the original values of the replicates. For example, if you have three measurements for the endo-1,4-b-glucanase activity at 24 h, the activity can be presented as a mean value and a standard deviation of the three measurements. You could explain what this SE stands for by using a superscript under the table.
L367: To which compounds is the text reffered?
Author Response
Dear reviewer, once again we thank you for taking the time to review our manuscript. We are pleased to know that the corrections made are in accordance with your suggestions. However, we are proceeding to respond to the new comments you provide.
- L92-93: What is the final volume of the culture?
Thank you for pointing out this detail. It was clarified that the analysis of carbon mineralization was carried out on the solid fermentation of the substrate as presented in line 92. In addition, the volume (1 mL) of the minimum salt medium used was included, line 94.
- L112-114: What volume of the supernatant was used for the reaction mixture? What was the final volume of the reaction mixture?
Thank you for clarifying this point, as the manuscript only referred to the amount of enzyme extract used in the case of the determination of laccase activity as well as the final reaction volume for this determination. For these reasons, the paragraph was rewritten and the following was added:
Line 113-114 100 μl of enzymatic extract was used in each enzyme activity assay. A final reaction volume of 1 mL was used for these determinations in each method.
- L166: How the Carbon-Nitrogen ration differs among the media?
Line 170-174 The difference in carbon-nitrogen ratio between the media used is due to their composition, where the PDA medium only provides complex carbohydrates and mineral salts necessary for the development of the fungi, in the case of CYE, in addition to sucrose, sodium nitrate is provided and finally, in the medium with malt agar, nitrogen is provided by peptone.
- L200-206: It would be more appropriate to include this paragraph under Section 3.2.
Thanks for pointing this out, it was corrected as you suggested.
- L224: Ammend "potency" to "power"
Thanks for pointing this out, it was corrected as you suggested.
- L271-278: It would be more appropriate to include this paragraph under Section 3.4.
Thanks for pointing this out, it was corrected as you suggested.
- L289: "at 24 hours" instead "after 24 hours"
Thanks for pointing this out, it was corrected as you suggested.
- L297-299: Please rephrase the sentences. In the previous sentence it is stated that "They (exoglucanases) are not very active against crystalline cellulose" while later it is stated that exoglucanases "would then proceed towards the crystalline regions of the fiber", which is contradictory.
Thank you for pointing this out, we agree, the paragraph has been rewritten as it appears below.
Line 300-305 Regarding the activity of exoglucanases, the highest values ​​are obtained after 48 hours of fermentation. These enzymes gradually attack the cellulose molecules at the non-reducing ends, releasing cellobiose subunits. They are active against crystalline cellulose and present a highly cooperative synergistic action in the presence of endoglucanases. The latter attack the amorphous regions of the cellulose fibres, thus creating new sites for the exoglucanases to continue the degradation of the fibre.
- Tables 2-4: Please amend the title of the "SE and P" column as in Table 1. However, it is still not understood what does this standard error stand for. Usually, a standard deviation expresses the deviation of a mean value from the original values of the replicates. For example, if you have three measurements for the endo-1,4-b-glucanase activity at 24 h, the activity can be presented as a mean value and a standard deviation of the three measurements. You could explain what this SE stands for by using a superscript under the table.
Thank you for pointing this out. The SE and P values ​​were corrected according to what is presented in Table 1.
Regarding the SE, this value corresponds to the standard error of the analysis of variance given by the statistical model. This model compares the enzymatic activity per day.
The enzymatic activity data for each day correspond to the mean value of three determinations, so Duncan's test is used to determine whether or not there are differences between the means, and the P value gives the level of significance of these differences
- L367: To which compounds is the text reffered?
Thank you for clarifying this aspect. In this case we are referring to the presence of sugars and aromatic compounds, so we decided to clarify it as it appears in line 375.
Reviewer 3 Report
The author has faithfully reviewed and supplemented the content suggested in the previous review. I believe that the references have been added and the content has been supplemented, making it a good document for readers to understand.
This paper can be published without any additional revision.
Author Response
Dear reviewer, once again we thank you for taking the time to review our article. We are pleased that the changes made are in accordance with your suggestions.
Round 3
Reviewer 1 Report
I have no other comments.
I have no other comments.